# Is Continuous Heart Rate Monitoring of Livestock a Dream or Is It Realistic? A Review

**DOI:** 10.3390/s20082291

**Published:** 2020-04-17

**Authors:** Luwei Nie, Daniel Berckmans, Chaoyuan Wang, Baoming Li

**Affiliations:** 1Department of Agricultural Structure and Bioenvironmental Engineering, College of Water Resources and Civil Engineering, China Agricultural University, Beijing 100083, China; nlwcau@cau.edu.cn (L.N.); libm@cau.edu.cn (B.L.); 2Key Laboratory of Agricultural Engineering in Structure and Environment, Ministry of Agriculture and Rural Affairs, Beijing 100083, China; 3M3-BIORES KU Leuven, Department BioSystems, Kasteelpark Arenberg 30, 3001 Leuven, Belgium; daniel.berckmans@kuleuven.be

**Keywords:** photoplethysmography (PPG), electrocardiography (ECG), photoplethysmographic imaging (PPGI), precision livestock farming (PLF), heart rate monitoring, livestock

## Abstract

For all homoeothermic living organisms, heart rate (HR) is a core variable to control the metabolic energy production in the body, which is crucial to realize essential bodily functions. Consequently, HR monitoring is becoming increasingly important in research of farm animals, not only for production efficiency, but also for animal welfare. Real-time HR monitoring for humans has become feasible though there are still shortcomings for continuously accurate measuring. This paper is an effort to estimate whether it is realistic to get a continuous HR sensor for livestock that can be used for long term monitoring. The review provides the reported techniques to monitor HR of living organisms by emphasizing their principles, advantages, and drawbacks. Various properties and capabilities of these techniques are compared to check the potential to transfer the mostly adequate sensor technology of humans to livestock in term of application. Based upon this review, we conclude that the photoplethysmographic (PPG) technique seems feasible for implementation in livestock. Therefore, we present the contributions to overcome challenges to evolve to better solutions. Our study indicates that it is realistic today to develop a PPG sensor able to be integrated into an ear tag for mid-sized and larger farm animals for continuously and accurately monitoring their HRs.

## 1. Introduction

The world needs livestock products to feed all people, and the total meat production was over 342.4 million tons in 2018 [1]. The Food and Agriculture Organization of the United Nations (FAO) estimates that the worldwide meat consumption may increase to 73% by 2050 [2], thus the food production, animal industry in particular, must become more sustainable. Currently, precision livestock farming (PLF) is regarded as the heart of the biological engineering endeavor towards sustainability in food production, using image and sound analysis, sensors, information technology, and decision-making to monitor, model, and manage animal production, reproduction, health, welfare, and environmental impact. Europe is considered the birthplace of PLF research, and it still continues strongly with over three decades of research and innovation through at least 4 EU-funded (EU-PLF, BioBusiness, AllSmartPigs, BrightAnimal) and many other national projects [3]. Current agricultural research agendas in the EU [4] and US [5] have evidenced that the importance of PLF is growing worldwide. Faced with the large worldwide demand for animal products, the question becomes: how many of these animals have a life worth living? This high number of animals is an opportunity to create sensors and hardware that can be very cheap per unit so they change the efficiency of the livestock sector and the animal welfare as described in many papers on PLF.

All humans and homoeothermic animals generate metabolic energy to live and to reproduce. For over 95% of their life, most of these living organisms generate their energy in the aerobic mode, by breathing air to lungs and by heart beats transporting the oxygen rich blood to the cells to produce metabolic energy. For homoeothermic living organisms, the heart rate (HR) is a crucial variable to control the metabolic energy production in the body by controlling the components in the metabolic energy balance. This includes the basal metabolism which refers to the minimum energy needed to keep all organs functioning in an extremely quiet state and thus to stay alive, the thermal component to control body temperature, the physical component, as well as the mental component, which is a key component in transferring feed energy efficiently into production and to prevent depression of the immune system due to stress. The less efficiently the metabolic energy is used in the body, the more feed energy will be wasted in manure, emissions, stress systems, etc. Therefore, HR is becoming increasingly important in research of farm animals, and so far it remains a challenge to monitor HR accurately and continuously by a reliable, affordable sensor on the animal or with a remote sensing technique. Current HR monitors for animals, such as implantable transmitters and externally-mounted equipment, are mainly used in research settings with the intentions of analyzing physiological responses, diseases, psychological and environmental stress, or individual characteristics, for instance the temperament and its coping strategies. They are however inconvenient and inappropriate for long-term continuous monitoring. In recent years, HR monitoring for humans has become feasible though there are still drawbacks in continuous and accurate measurement. In this paper, we present a comparative review of current techniques to measure HR on living organisms, with focus on their advantages and drawbacks, and discuss the potential to transfer some of the techniques that have been successfully applied in humans to livestock.

The objective of this paper is to check which techniques are feasible for the continuous HR monitoring on livestock based upon the described technologies presented in literature. Additionally, we estimate the challenges and propose a solution to get such sensors.

## 2. Physiological Effects

All HR monitoring methods, presented in this review, are aimed for real-time detection of HR. We focus on HR defined as the number of cardiac beats at a given moment, which is usually expressed in beats per minute (bpm). According to the principles of different techniques, acquisition of HR signals relies on specific physiological effects. Figure 1 provides an overview of different categories of physiological effects, and how they are linked to the HR, as well as the measuring techniques. Typically, such effects are comprised of bioelectrical effects, mechanical effects, and thermal effects [6].

Bioelectrical effects: Electrical excitation of the heart causes dynamic electromagnetic fields on the body surface that can be measured by electrocardiography (ECG), such as wet and dry electrodes, and capacitively coupled ECG (CCECG).

Mechanical effects: Blood travelling through the vascular system causes organ motion and deformation, as well as blood volume variation. These phenomena are mechanical operations and can be subcategorized into three groups as follows.

(1) Body surface displacement: At every heartbeat, the pulse wave travelling through the body produces subtle changes in displacements and vibrations of the body surface. Several sensor techniques rely on these effects, such as ballistocardiography (BCG)/seismocardiography (SCG), Doppler radar and lasers (optical vibrocardiography), as well as video-based motion.

(2) Superficial perfusion: The ejection of blood from the heart into the vascular tree causes blood volume changes in the microvascular bed of the tissue. Since blood absorbs light more than the surrounding tissues [7], these microscopic changes in the optical properties of the body surface can be measured by photoplethysmography (PPG) and photoplethysmographic imaging (PPGI) methods.

(3) Intrathoracic dynamics: The impedance distribution within the human body varies with physiological activity. During the cardiac cycle, the motion of the cardiac wall and aorta, as well as the opening and closing of heart valves, causes variations of impedance distribution. Cardiac pulsation also modulates tissue impedance by blood perfusion [8]. The local changes in impedance caused by the cardiac cycle inside the thorax do not project actively onto the body surface [6]. However, the electrical/magnetic impedance measurements can be used to detect the variations.

Thermal effects: The flow of blood through the vicinity of major superficial vessels leads to changes in skin temperature that can be detected using thermal imaging techniques.

## 3. Available Techniques to Monitor HR

This section introduces the various state-of-the-art techniques for HR monitoring of living organisms with their advantages and limitations.

### 3.1. Electrocardiography (ECG)

The conventional ECG method is considered to be one of the oldest optimal clinical diagnostic tools with its first recordings dating back as early as 1903 [9]. It provides useful information about the cardiovascular system, characterized by its high accuracy and easy interpretation for HR measuring. The method employs Ag/AgCl electrodes with wet conductive gels fixed to specific locations of the chest, arms, or hands and legs in order to detect and record the difference of the electric potential between the points. This electrical excitation generated by cardiac muscular fibers causes a voltage signal that allows accurate measurement of HR. Although the conventional ECG provides good signal quality, it is inconvenient and inappropriate for long-term monitoring (>1 week), since the wet electrodes need to be directly contacted with skin. It also presents movement limitations, and prolonged application of wet electrodes may cause skin irritation, allergic reactions, and signal degradation due to dehydration [10].

In order to overcome the drawbacks, dry electrodes [11,12] and capacitively coupled ECG (CCECG) [13,14] are two alternatives which potentially provide comfortable measurement without an explicit electrolyte. However, the dry electrodes still need to be in direct contact with skin, thereby it may cause skin irritation and allergies after prolonged use. CCECG is a non-contact method, which is able to detect biopotentials with an explicit gap (e.g., a thin layer of insulator) between electrodes and skin, even through hair and clothing. Polar Sport Tester uses belt monitors, which are primarily marketed for sports and relevant research in sports medicine, and fabricated by a commercial manufacturer (Polar Electro Oy, Kempele, Finland) to record inter-beat intervals (IBIs). Some recent examples integrated the CCECG sensor into a chair [15,16], bed [17], belt [13], or clothing [14] to obtain ECG signals. Compared to the wet and dry electrodes, the surface of CCECG electrodes is electrically insulated and thus remains stable even in longer-term usage.

Several authors have demonstrated the performance of the dry/non-contact electrodes, which are comparable to clinical wet electrodes as shown in Table 1, including the implementation of sensors, frequency of movement of subjects when measuring, power consumption, and quantitative results. However, a tight vest and chest band is often needed to secure the non-contact electrodes in place in the form of textile electrodes [18]. Although the CCECG measurement can detect ECG signals accurately, it has several shortcomings. Due to the high impedance, the signal quality is not comparable to conventional wet electrodes. Moreover, it is susceptibly affected by motion artifacts (MAs). Any changes in the displacement of the electrode-to-skin distance can change the coupling capacitance, hence affecting the ECG signal acquired, and friction between the electrodes and insulation may also cause artifacts.

### 3.2. Photoplethysmography (PPG)

Another widely used conventional technique for HR measurement is photoplethysmography (PPG), which was first introduced by Hertzman in 1938 [26]. Its principle has been reviewed previously [27]. PPG is a simple, non-invasive optical measurement technique that detects blood volume changes in the microvascular bed of tissue [28]. It is based on the principle that blood absorbs more light than surrounding tissues [7]. The measurement is done by illuminating human tissue (skin) with a light source, which is usually a red, near infrared (NIR) [9], or green light, and a photodetector to opto-electronically detect variations in the intensity of transmitted or reflected light. The changes in light intensity are associated with small variations in blood perfusion of the tissue and provide information on HR.

In general, PPG can be operated in a transmission or a reflection mode [29]. In transmission mode, the light transmitted through the medium is detected by a photodetector opposite to the light source, in which a relatively good signal can be obtained. But it is restricted to certain thin areas (earlobe and fingertip) of the body, where the signal can be quickly detected. A typical application is the conventional pulse oximetry sensor, while for such a system the fingertip sensor may interfere with daily activities. Commercial ear-clip PPG sensors can cause pain over long-term use. In the reflection mode, the photodetector detects the light that is back-scattered or reflected from tissue, which reduces the problems associated with sensor placement. It can be used on a variety of measurement sites, which offer higher perfusion values, including finger [30,31], wrist, earlobe [32], external ear cartilage [33], superior auricle [34], inferior auricle [35], forehead [36], and brachia [37].

Such devices are commonly incorporated with ear-worn devices [33,34,38], finger probes [39], flexible films or patches [40,41], and glass-type wireless PPG [42]. Table 2 and Table 3 provide detailed comparisons of some significant contributions for HR monitoring on a portion of the above-mentioned measurement sites (finger and ear), including light wavelength, movement of the subjects when measuring, power consumption, and quantitative results (accuracy), which are compared to gold standard methods (conventional PPG or ECG). These results indicate that PPG signals recorded from fingers and ears are acceptable for HR monitoring, but most of them focused on slowly running, which considers low MAs. Only a few techniques were proposed for HR monitoring in fitness, which generate strong (high amplitude) MAs.

In recent years, wearable fitness trackers and body sensor devices based on PPG have become increasingly popular. More companies are producing these sensors. A review is provided evaluating the accuracy, precision, and overall performance of wearable devices currently available [48]. The results indicate that these devices are relatively accurate and might be beneficial. In addition, wrist-wearable devices like Xiaomi (Xiaomi Corporation, Beijing, China) and Fitbit (Fitbit, Inc., San Francisco, California, USA) have recently received considerable interests [49] due to their low prices, as in case of the Xiaomi Mi Band priced at $14 [50].

Compared to fingers and earlobes, due to large flexibility of the wrist and loose interface between the sensor and the skin, the quality of the wrist-type PPG sensor signal is susceptible to MAs during intensive physical exercise. Therefore, it is challenging as well as of great interest to remove MAs and to accurately estimate HR for wrist-type wearable devices. Many contributions have shown improvement in the accuracy of HR measurement using wrist-type PPG.

Thanks to the IEEE Signal Processing Cup 2015 [51], the database which was recorded from 12 healthy subjects walking or running on a treadmill with varying speed ranging from 6–8 km/h to 12–15 km/h is publicly available. Each dataset consists of two-channel PPG signals, three-channel acceleration signals, and a single-channel ECG signal recorded by using reflective pulse oximeters with green LEDs (wavelengths of 515 nm or 609 nm), a three-axis accelerometer, and wet ECG sensors on the chest as the ground-truth of HR, respectively. Both the pulse oximeters and the accelerometer are embedded in a wristband. It should be noted that utilizing the same dataset and performance metrics which are defined facilitates the comparison between the results achieved with different state-of-the-art approaches. Several HR estimation methods have been designed and tested, and detailed summaries containing signal processing methods and quantitative results of significant studies are presented in Table 4 and Table 5.

To evaluate the performance of the proposed methods, multiple measurement indexes were adopted in these studies to analyze the relationship between estimates ($BPM_{est}\left(i\right)$) and ground truth $\left(BPM_{true}\left(i\right)\right)$ values [52]. One was the average absolute error (*Error 1*), defined as:(1)$$Error\;1=\frac{1}{W}\overset{W}{\underset{i=1}{\sum }}\left|BPM_{est}\left(i\right)-BPM_{true}\left(i\right)\right|$$
where *W* is the total number of time windows. Similarly, the average absolute error percentage (*Error 2*) was calculated, defined as:(2)$$Error\;2=\frac{1}{W}\overset{W}{\underset{i=1}{\sum }}\frac{\left|BPM_{est}\left(i\right)-BPM_{true}\left(i\right)\right|}{BPM_{true}\left(i\right)}$$

The Bland-Altman plot [53], based on graphical techniques and simple calculations, directly reflects the agreement between ground truth values and estimates. In this analysis, the limits of agreement (*LOA*) is used, which is expressed as LOA=[μ−1.96σ,μ+1.96σ], where μ, σ denote the average difference and the standard deviation respectively.

### 3.3. Photoplethysmographic Imaging (PPGI)

Verkruysse et al. (2008) first introduced photoplethysmographic imaging (PPGI) using ambient light [65]. PPGI is the translation of PPG into a spatially resolved, non-contact method [66]. It is based on the conventional PPG theory, i.e., the skin changes its optical properties with perfusion. Compared to PPG, the PPGI replaces the photodiode, which should be in contact with the subject’s skin in a single location, with cameras that can be found everywhere nowadays, such as digital camera [67], webcams [68], and cellphone cameras [69] with dedicated light sources (e.g., green, red, and/or IR wavelengths) or normal ambient light. Such camera systems monitor a larger field of view of the subjects’ exposed skin (usually facial area) from a distance, to record small changes in light intensity values from skin and to capture HR over time.

Compared to conventional PPG technique, as a contactless technique, PPGI avoids the requirement of the device to be in contact with the skin and the deformation of the arterial wall. Beyond that, PPGI also has several strengths: (1) cameras (particularly webcams) are ubiquitous and often inexpensive; (2) it offers detailed spatial information simultaneously from multiple sites of arbitrary size and location [70]; and (3) measurements from multiple subjects can be performed [71]. One sensor can monitor more individuals, which is economically interesting. The application of PPGI has evolved since 2010, and many contributions have been proposed to improve the performance of PPGI signal. The existing review [46] provides theoretical background and an overview of the field. A summary has been outlined in Table 6, Table 7 and Table 8, containing sensor, experimental conditions (the illumination, the distance between subject and camera, and the frequency of movement of subject when measuring), signal processing techniques, and accuracy.

However, the acquired signal of PPGI is highly susceptible to motion-induced signal corruption, and the MAs removal or attenuation is one of the challenges in signal extraction and processing. The illumination variations, i.e., low light levels and dynamic variations, and skin pigmentation could contaminate the pulse signal. The drawbacks limit the physiological monitoring capability of the technique in practice so far. A variety of signal processing methods have been proposed to remove noise and detect HR from the PPGI signals. Robust image and signal processing methods have the potential to address the MAs and make it possible to have a good performance for HR monitoring.

### 3.4. Video-Based Motion

The principle of extracting HR from facial video is measuring subtle head motion caused by blood flow from the heart to the head that is not visible by naked eyes, which is based on BCG theory. The blood circulation from the heart to the head via the carotid arteries causes the head to move in a periodic motion. Wu (2012) developed a method called as Eulerian Video Magnification to amplify these subtle changes [86].

Balakrishnan et al. (2013) extracted the HR from the facial video by tracking velocities of feature points on the face region [87]. The average HRs were closely identical to the true ones for all subjects. The mean error of average HRs was 1.5% compared to a wearable ECG monitor. Compared to PPGI, the BCG based video technique has several advantages. Since the method detects the motion of feature points on regions of interest (ROI), it is invariant to illumination and reliable even when the face is occluded. However, the approach is highly vulnerable to motion variations of subjects. Thus, researchers proposed that combining the features of the BCG based approach and PPG based approach could overcome the limitations of the methods related to motion variance and illumination variance [88].

### 3.5. Thermal Imaging

The human skin surface with a temperature of about 300 K emits electromagnetic radiation in the far infrared part of the spectrum, which is not visible to the naked human eye. The human skin temperature in the vicinity of major superficial vessels is directly modulated by the pulse blood flow. The pulse signal containing the HR can be recovered from the subtle changes in skin temperature recorded with a highly sensitive thermal camera at a distance and processed through appropriate signal analysis. The approach was tested on motionless human subjects up to 3 to 10 feet away from a mid-wave infrared camera to measure HR from the wrist, neck, and forehead area [89]. Results achieved 98% agreement and mean bias was 4.74 ± 9.28 bpm of HR compared to the reference signal from a piezoelectrical transducer.

### 3.6. Ballistocardiography (BCG) and Seismocardiography (SCG)

At every heartbeat, the blood travels along the vascular tree and produces subtle changes in displacements and vibrations of the body surface. The BCG and the SCG techniques both record different aspects of the mechanical activity of the body. BCG measures whole-body recoil forces in response to blood ejection into the vascular tree, while SCG detects the positional vibrations of the chest wall in reaction to the myocardial motions [90]. In principle, the recordings of the BCG and SCG contain useful physiological information, such as HR.

Although BCG and SCG measure different aspects of cardiac activity, many sensor techniques actually record a superposition of both signal sources [91]. Thus, both are jointly considered for signal analysis and processing. Table 9 summarizes the BCG and SCG measurement for HR monitoring, and these mechanical methods require physical coupling to the body surface and use force sensors [92,93], pressure sensors [94,95], film sensors [96], stain gauges [97,98], optical sensors [96,99,100], and acceleration sensors [101,102]. As they are not required to be directly attached to the body surface, the sensors can be integrated into furniture, such as beds, chairs, and weighing scales. Thus, the mechanical assessment of human HR allows an unobtrusive measurement.

### 3.7. Doppler Radar and Laser

During each cardiac cycle, the heart muscle pumps blood through the circulatory system, which results in displacements and vibrations of the body surface, e.g., chest wall. The phenomena can be measured by using Doppler radar and laser techniques, which are based on the Doppler effect with certain displacement resolutions from a distance.

Radar-based approaches make use of electromagnetic radiation to measure body surface motion related to heart mechanical activity in a noncontact manner, so it is possible to remotely sense HR. In the past decades, there were three types of radar systems proposed for HR monitoring, namely constant-frequency continuous wave (CW), frequency-modulated continuous wave (FMCW), and ultrawideband (UWB) [6]. According to the Doppler theory, a target with a time-varying position reflects the transmitted signal with its phase modulated proportional to the target displacement [106]. Continuous wave is the most commonly used method to detect HR among the aforementioned three types, and it is based on the frequency shift. When CW measures the velocity of target, the phase detecting radar adopts the method of analyzing phase differences between transmitted and received signals to measure the distance to a target. Due to its fine resolution, it has been employed in HR monitoring.

Another non-contact technique, which measures the displacement of body surface such as the chest from a distance, utilizes a laser Doppler vibrometer. The approach is referred to as optical vibrocardiography. With high displacement resolution, the laser has the ability to detect subtle deflections of body area caused by the heart activity by measuring the frequency or phase differences between a reference beam and a test beam [107]. Table 10 shows related research for HR measuring using Doppler radar and laser techniques, including sensors, the distance between the sensor and subject, the movement of the subject, and quantitative results.

### 3.8. Impedance

During cardiac cycle, due to motion of the thorax and of inner organs, the thoracic conductivity distribution varies, which causes variation in the impedance. The electromagnetic coupling which includes methods using inductive and capacitive coupling can be used to enable noncontact measurement of these variations [8]. The magnetic induction method is based on coils sending out an alternating magnetic field, which induces eddy currents in the conductive body. The eddy currents generate an opposing magnetic field; therefore, a new effective coil impedance is established, which is modulated by changes of the conductivity inside the body. When using the electrical impedance for recording physiological activity, the electrical field is capacitively coupled into the body. The method is based on the fact that two metal plates together with the body form a capacitance. Due to the magnitude and direction of the induced displacement, currents vary with the electric property distribution of the body, and the capacitively received signal reflects the changes of tissue parameters.

### 3.9. Animal HR Monitoring Techniques

HR is a crucial variable to quantify animal welfare and health state, process efficiency, and environmental impact. A real-time and continuous monitoring of HR of animals is a potential tool to improve its production efficiency and to monitor animal welfare based upon objective physiological signals [115]. Animal HR monitoring methods currently reported in literature are mainly based on acquiring ECG signals or IBIs directly because of the similarities in the heart characteristics of humans and animals such as pigs, which are excellent models of human cardiovascular disease [116]. Various portable devices are commercially available to record ECG signals for animals. Holter systems, widely used in human clinics, are sometimes employed for ambulatory long-term recording of ECG signal (mostly up to 24 h) for animals [117]. It is not practical to apply them to livestock, since these systems are very expensive and technically vulnerable. Another relatively cheaper measurement is to use belt monitors, which are primarily marketed for sports and relevant research in sports medicine, and fabricated by a commercial manufacturer (Polar Electro Oy, Kempele, Finland) to record IBIs data, like Polar Sport Tester. Polar S810i, Polar Vantage NV, and Polar R-R Recorder are widely applied in veterinary and behavioral research, all using an electrode belt containing two coated electrodes that fit around the thorax of the animals. When applying such wearable techniques to livestock, it is inconvenient to mount it on animals tightly, and the obtrusive devices are also easily discovered by their mates, which may cause aggressive behaviors. Implantable telemetric transmitters, which have been applied in a wide range of small size laboratory animals, address the drawbacks mentioned above, but it is an invasive approach to measure HR, since surgery or injection is needed, thereby animals need a couple of days of recovery after the procedure. Furthermore, complications during the procedure may emerge.

In recent years, a variety of less invasive and attached methods have been proposed to assess HR in farm animals. The PPG based sensor located in the ear of pigs can provide HR information. Due to the motion-induced artifacts, the quality of signal is not good enough accurately measure the HR of animals [118]. Researchers are currently investigating non-contact computer-based techniques to monitor HR in livestock [119]. A low to high correlation coefficient (0.09–0.99) was found between HR obtained from the red, green and blue color channels (RGB) videos and from the gold standard method in cattle [120]. The study also showed that the accuracy of HR measurement is related to both the position of the cameras and the area analyzed on the images. Thus, it is necessary to develop an accurate and convenient promising approach for real-time continuous animal HR monitoring.

## 4. Transferable HR Monitoring Techniques in Livestock

### 4.1. Evaluation Criteria

The techniques discussed in Section 3 are reported to be successful and capable of providing HR measurement with a certain reliability and accuracy for humans, which provides new ideas to transfer them in livestock. Each of the techniques has its advantages and limitations that are method dependent. These techniques have different performances in specific environments. The comparison of various properties and capabilities related to application in livestock are discussed, including the measuring sensor, the distance between the subject and sensor, the frequency of movement of the subject when measuring, and the cost, which are shown in Table 11. Then we evaluate the feasibility of each technique for application to livestock.

• Accuracy

Accuracy is an essential factor for HR measurement. A total error within 5 bpm is an acceptable margin [72]. According to the discussion above, it can be seen that almost all the techniques are reliable for HR monitoring in a specific scenario.

• Distance

Distance between the sensor and the animal is an important variable for the assessment of the different techniques and their potential applications in livestock, since accurate measurement does not only depend on the technique itself, but also on the specific implementation. Table 11 shows the orders of magnitude for distances. According to our definition of the categories of wet/dry electrode methods, they require direct contact with the subjects. Gaps between the sensor and subject in the range of millimeters are required by PPG, BCG/SCG, and CCECG methods. For the impedance technique, slightly increased separation of electrodes or coils and subject is possible, and the distances can be in the range of centimeters [6]. Significantly longer distances can be covered by the remaining systems, which are camera-based techniques like PPGI, thermal imaging, and video-based motion, as well as radar and lasers. Such systems can be easily operated from distances in the range of meters.

• Movement

It can be seen that all investigated HR monitoring techniques are to some extent sensitive to MAs. Hence most of them require that the detected subjects remain motionless. For wearable PPG and CCECG technologies, even though they have good performance even in extensive exercises, close proximity to the body surfaces is also necessary.

• Cost

Cost of the different techniques is fuzzy if no limiting definitions are employed. Here we just consider the lowest possible cost for sensors. The cost of algorithm development, which is a very important aspect for the commercial success of a technology, is beyond the scope of this review. Considering the measuring sensors for various techniques, the techniques PPG, PPGI, video-based motion, and the BCG/SCG approach exhibit the lowest costs. The PPG system consists of inexpensive phototransistors. Video-based motion and PPGI systems can be realized using consumer grade cameras and webcams that are mass produced; hence they are very inexpensive as well. The sensors used for BCG/SCG methods, such as very simple optical sensors, strain gauges, and pressure sensors, are very cheap. We consider the techniques of ECG, impedance, and radar to be in the medium range of costs. Although the principle of CCECG is simple, high demands of the design and assembly process are needed to achieve ultrahigh-input impedances. For the impedance method, complex components like coils or electrodes and assembly processes are necessary. The laser is an expensive method because of its complex optical assemblies and costly devices. The most expensive method among all of the techniques is thermal imaging because a very sensitive thermal imaging camera is needed when monitoring perfusion.

### 4.2. Transferable Feasibility from Human to Livestock of Various Techniques

• ECG

The wet and dry electrodes both require direct contact with the skin of measured subjects. The attachment site should be cleaned, and its hair should be shaved off if necessary. Wet electrodes need conductive gel, while dry electrodes depend on sweat and moisture between the electrodes and skin, which may cause skin irritation and allergic reaction after prolonged usage. Although CCECG is a non-contact method to detect ECG signal even through hair and clothing, it is sensitive to corruption by MAs. In order to secure the non-contact electrodes in place, a tight vest, chest band, or strap are required. When applied to livestock, it is inconvenient to be mounted on animals, and the obtrusive devices are easily discovered by conspecifics, which may cause aggressive behaviors.

• PPG

The contact PPG is a low cost, simple, and non-invasive optical measurement, which provides good accuracy of HR measurement for humans even during intensive physical exercises. It was concluded that PPG technique has the potential for a transfer to livestock. Previous research has identified several factors that affect PPG signals, and it is still challenging to develop such a wearable PPG sensor for livestock. The underlying factors are discussed in Section 5.

• PPGI

Compared to the contact requirements of PPG, the PPGI is based on the camera, which is a non-contact, low-cost, and convenient technique, and is preferable above a PPG sensor for the reasons of hygiene, animal welfare, and practical installation for housed animals. From literature we can conclude that although these studies showed promising results, most of them require that subjects remain motionless and facing the cameras during recording. The distance between the camera and measured subject within several meters is another limit. PPGI signals are susceptible to motion-induced artifacts and illumination variations, particularly when dealing with webcams during ambient light [76]. Due to the complex environment of livestock, including low light levels and variations, movement of animals, and their distance to cameras, many issues of the signal processing techniques remain to be addressed to transfer PPGI to livestock in the future.

• Video-based motion

The video-based motion technique for HR measuring possesses certain advantages over the PPGI methods. For example, it is invariant to illumination variance and not restricted to any particular view of the face. Furthermore, it is effective even when skin is not visible, while for PPGI, the sensitivity to color noise and changes in illumination should be considered, which requires facial skin to be exposed to the camera during tracking. However, the video-based motion approach would be highly vulnerable to internal facial motion and external movement of the head [121]. In realistic scenarios, particularly in livestock farming, it is still a big challenge to use current methods to obtain high-quality video due to the unavoidable voluntary motion variations of the animals.

• Thermal imaging

Thermal imaging is a passive and non-contact method, and still presents several limitations. Its signal is affected by physiological and environmental thermal phenomena [122]. Results so far come from research in experimental controlled settings, where the subject remains motionless. Any spontaneous movements, such as small movements of the limbs or even stressed breathing, affect the shape of the measured signal dramatically. Furthermore, the method is dependent on unwanted thermal distortions, such as thermal exchanges, sweating, external heat radiation, airflow, etc. Moreover, the infrared-based measurement is much more expensive, as high-quality cameras must be used. We don’t think it is currently practical to employ the technique particularly at the farm level, due to the uncontrolled animals, environment, and the costly cameras.

• BCG and SCG

BCG and SCG techniques often show high signal-to-noise ratios, which allow extraction of HR information accurately from signals on the basis of quantitative results in Table 9. The signals are vulnerably affected by MAs caused by body movements, position, and respiratory movements, and hence the quality of the corresponding HR recordings is impacted [123]. We don’t think it is currently appropriate and applicable in livestock.

• Doppler radar and laser

With regard to Doppler effect techniques, the experimental protocols in related research require the subjects to remain motionless to limit body movements, since the Doppler radar and lasers, which have high displacement resolution, are prone to be affected by MAs, and even a small movement of the body can result in a high deformation of the signals. In addition, Doppler effect techniques are active measuring methods since they emit energy. When applying Doppler in livestock, the direct contact of the laser to the animals’ eyes is a risk for animal welfare. For Doppler radar, the greater the required measuring distance, the higher the frequency and transmitted power are needed, which are prone to present a safety threat. Furthermore, these technologies are relatively expensive because of the special hardware required.

• Impedance

When conducting electrical/magnetic impedance measurements with human subjects, as the changes in the heart are smaller than the lung, the signal content related to respiratory activity is always higher than the cardiac signal [8]. Thus, it requires accurate positioning of the sensor to optimize the monitoring related to more cardiac information [124]. Moreover, impedance measurement is vulnerable to motion. Cardiac activity modulates the impedance distribution by organ motion; thus, it is difficult to distinguish between signals related to cardiac activity and random motion noise. Therefore, even with a minor motion such as finger movement, the performance of HR monitoring will degrade [125]. In an ambulant environment, particularly in livestock farming, it is still a big challenge to use the technique due to the unavoidable voluntary motion variations of the animals.

According to the properties of various techniques, a low-cost, motion-resistant, and accurate sensor is needed for animal HR monitoring in livestock. We conclude that the PPG principle techniques might be the mostly transferable since they satisfy the above requirements.

## 5. Challenge and Future Work

To transfer the PPG technique successfully applied in human beings to livestock, one important consideration is whether the PPG theory based on skin blood perfusion is applicable for animals. Thus, the similarities of skin between humans and animals need to be checked first. For mammals, such as pigs, there are several anatomical and physiological similarities to humans, including skin characteristics. Porcine skin is increasingly being employed as a model of human skin in various research fields, including pharmacology, toxicology, and immunology, with particular interest in percutaneous permeation and organ transplantation for wound healing. There are several anatomical and physiological similarities between porcine and human skin. As in humans, porcine skin is divided into three layers, i.e., the epidermis, the dermis, and the hypodermis (or subcutis) from top to bottom. Human epidermis ranges from 50 to 120 μm and pig epidermis ranges from 30 to 140 μm. The epidermal thickness varies considerably based on body site, and a better measure is the dermal-epidermal thickness ratio [126]. It has been reported that this ratio ranges from 10:1 to 13:1 for pigs, which is comparable to human skin. Moreover, the size, orientation, and distribution of blood vessels in the dermis of pigs are also similar to blood vessels of human skin. Additionally, the dorsal site of pig ears represents the area of porcine tissue with the highest similarity to human skin, with regard to the thickness of the different skin layers [127].

From the above, we conclude that PPG theory has the potential for HR assessment for pigs, since porcine skin is known to be similar to human skin, anatomically and physiologically. To design such a wearable PPG sensor for livestock, several factors that affect PPG signals need to be considered, including motion artifacts removal, measurement site, light wavelength, contact force, power consumption, weight and size, and the expected price. This section briefly discusses these factors.

• Motion artifacts removal

Despite the attractive attributes of PPG, a major limitation is that PPG is sensitive and vulnerable to MAs. The quality of PPG signals can be easily affected by movement especially during intensive physical exercise. MAs on signals are considered the relationship between motion and noise. This includes voluntary and involuntary movements of the interface between the sensor and skin (tissue) [128]. The change in blood flow due to movements is another MA source [37]. In fact, it is difficult to remove MAs because they do not have a predefined narrow frequency band and their spectrum often overlaps the PPG signals [129]. Consequently, MA removal in the original PPG signals is a challenging task.

To date, many signal processing techniques have been proposed to remove MAs in PPG signals. Adaptive noise cancellation (ANC) is a popular approach to remove MAs where reference signals can be constructed from acceleration data or another PPG signal [39,120,121,122,123,124,125,126,127,128,129,130,131,132,133]. However, the drawback is that the performance of ANC is sensitive to the reference signal, and it is difficult to reconstruct qualified reference signals during exercising. Independent component analysis (ICA) is another technique to remove MAs. Kim and Yoo (2006) proposed a basic ICA algorithm and block interleaving to remove MAs [134]. Krishnan et al. (2010) suggested using frequency-domain-based ICA [135]. However, the key assumption of ICA is statistical independence or noncorrelation; MAs contaminated PPG signals are not satisfactory [136].

Other MA removal techniques include the wavelet-based method [137,138,139], non-line methods [140], empirical mode decomposition [141,142,143], time-frequency analysis [144], Kalman filtering [145], electronic processing methodology [146], and spectrum subtraction using acceleration data [147], to name a few. However, most of the above techniques were proposed for small motions, where MAs were not strong.

Motion is often an arbitrary and spontaneous behavior which can be rhythmic or non-rhythmic. The acquired PPG signals exhibit both MA spectral peaks that lie distant from HR spectral peak as well as overlapping ones, rendering them indistinguishable [148]. In the presence of rhythmic movements such as running and walking, the peak corresponding to MA might be close to the HR peak. With regards to this situation, a number of signal processing techniques have been proposed for robust HR estimation from the PPG signals and acceleration data when subjects are running on treadmills (see Table 4 and Table 5). The MA cancellation and spectral peak tracking for HR extraction are both important to gain improvements in accuracy and robustness of HR estimation. However, non-rhythmic exercises, like boxing, can give large erroneous peaks if the entire window is considered. Sometimes, estimating HR during steady running is easier than the rest position, since the MA peaks are large in number and are scattered all over the spectrum in the latter case [61]. For ambulant livestock conditions, the movements of animals are generally non-rhythmic, and it is more challenging to MA reduction algorithms. However, the promising techniques for humans can be used as references in the future to deal with the PPG signal in livestock. As solutions and patents already show, there are techniques to reduce the MA regardless of whether they are rhythmic or non-rhythmic as shown in at least eight patents [149].

• Measurement position

Although sensing components are physically changed to decrease MAs, more analysis is needed to determine which measurement location is the best for HR monitoring. For humans, the perfusion values of 52 anatomical sites showed that the fingers, palm, face, and ears offer much higher perfusion values compared with other locations, and that especially the earlobe provides the largest perfusion values in transmitted PPG signal [150]. Likewise, porcine ear was chosen because of its higher cutaneous perfusion, lower body fat, and more suitable to place the sensor probe in practice. The development of ear tag for pigs is basically complete and practical. We consider that it is realistic to integrate the PPG technique into an ear tag today.

• Light wavelength

The wavelength determines the penetration depth of light [151]. Infrared and near infrared lights can pass easily and measure the deep-tissue blood flow because of their longer wavelengths. The shorter wavelength of green light will not reach deeper tissues, but it is suitable to measure the superficial blood flow in skin. Thus, for transmission mode, green light does not suit. As a result, the infrared wavelength is often used as the light source. Compared to infrared light, green light has much greater absorption for both oxyhemoglobin and hemoglobin. Reflected green light is less influenced by the tissue and vein region, resulting in a better signal quality [152]. That is why green wavelength PPG devices are becoming increasingly popular in recent years. Signal quality is largely influenced by the skin properties like skin pigmentation [153]. A longer wavelength yellow-orange light showed consistent improvement in the signal quality of PPG acquired from varied skin tone subjects [154]. For livestock, suitable mode (transmission or reflection) and light wavelength for porcine ear skin both need to be tailored to maximize the signal quality acquired.

• Contact pressure

The PPG signal waveform may be affected by the contacting force between the sensor and the measurement site. Insufficient pressure may cause inadequate contact and result in low PPG signal amplitude. On the other hand, excessive pressure conditions may also lead to low PPG signal amplitude because of blood occlusion. As a result, pressuring the tissue increases the pulsation amplitude and thereby provides a better signal to noise ratio, but then decreases with increasing contact force [155]. Although there are no generally accepted standards for contact pressure, some attempts to define it have been conducted. Daniel et al. (1981) suggested the pressure less than 90 mm·Hg causes no long-term circulation problems [156]. For different contacting forces, from 0.2 N to 1.0 N, most of the subjects achieved their maximum pulse amplitude within 0.2–0.4 N [157]. Another work applied 0–200 mm·Hg contact pressure to the finger, and indicated that the highest amplitude ratio was 60 mm·Hg [158]. Rhee et al. (2001) examined the relationship between skin pressure and pulsation amplitude using finger ring sensors [43]. The experiments indicated that the pulsation amplitude increased until the skin pressure reached 100 mm·Hg, and then gradually decreased. At around 180 mm·Hg, the blood completely occluded and thereby the pulsation disappeared. The results suggest that the effects of contacting force should be carefully examined in the design of a PPG ear tag.

• Power consumption

Since the PPG ear tag uses batteries, it requires small batteries to fit the tag size. To some extent, the battery size determines its capacity, so a very low power consumption sensor is demanded. Little research presented battery lifetime. The total current consumption of a finger-ring PPG sensor is 0.491 mA, and the transmitter and CPU-LED circuit consume 0.098 mA and 0.393 mA, respectively. The ring sensor can continuously detect and transmit PPG signals for 23.3 days, while the battery life can be extended to several months with an intermittent measurement schedule [43]. Some algorithms work well for data sampled at a low sampling rate, thus saving energy consumption in data acquisition and in wireless transmission [55,60]. To extend the battery lifetime, all components of the ear tag sensor should be chosen from low-power options, and some battery-saving techniques are also necessary. Particularly, data communication is always one of the most power-hungry parts, thus an extremely low-power wireless communication protocol should be used.

• Weight and size

As mentioned previously, the weight and size of the pulse oximeter is small enough to be integrated in a finger ring, ear-worn device, and wristband. In animal ecology studies, in order to minimize the effects of the tracking devices on animal behavior and performance, the weight of body-mounted equipment is commonly recommended to be less than 5% of the animal’s body weight [159]. The size of current Radio Frequency Identification (RFID) ear tag for individual identification is appropriate for PPG sensors. We conclude that this sensor technology, being able to be integrated into an ear tag, can be employed for mid-sized and larger animals, and the application should be started with cow, pig, sheep, goat, etc. Within the coming years, we expect more miniaturized hardware, making the concept also applicable to smaller animals, such as poultry.

• Cost

For the price of the sensor, the PPG is a low-cost technique. When a large number of sensor units are manufactured, reduced price is expected. This means that the big industrial farms, using high numbers of sensors, will help to reduce the price so that small farms can also afford using these products.

## 6. Conclusions

Knowing that this year again over 65 billion animals will be slaughtered for food production, it would be a serious advantage if we could monitor animal health and welfare in a continuous and automated way. This needs accurate, reliable, and affordable sensors. As we have seen so far, there are currently a significant number of different sensor technologies for HR monitoring under investigation, and this paper has critically reviewed the progress. Eight promising measuring technologies used on human beings were investigated, and the principles and the theories of HR measurement were discussed. Moreover, advantages and drawbacks with further elaboration were emphasized by comparing various properties and capabilities related to the application to livestock. By analyzing the challenges to design such a sensor, we conclude that it is realistic today to develop a continuous PPG sensor for HR monitoring that can be integrated into an ear tag for mid-sized and larger animals, such as cow, pig, sheep, goat, etc. Research endeavors of HR online monitoring on pigs would be a game changing milestone in the livestock sector. Moreover, the monitoring could become applicable for small species such as poultry when more miniaturized hardware is realized in a predictable future. We hope that this study will inspire researchers and technology companies to invest in such technology and develop prototypes of the ear tag as well as to produce the sensors in very high numbers. It will allow monitoring of animal welfare based upon physiological variables and to follow the metabolic energy balance of animals day and night in an automated way. This energy balance directly links to production results and creates disruptive innovation for animal health monitoring.

## Figures and Tables

**Figure 1 sensors-20-02291-f001:**
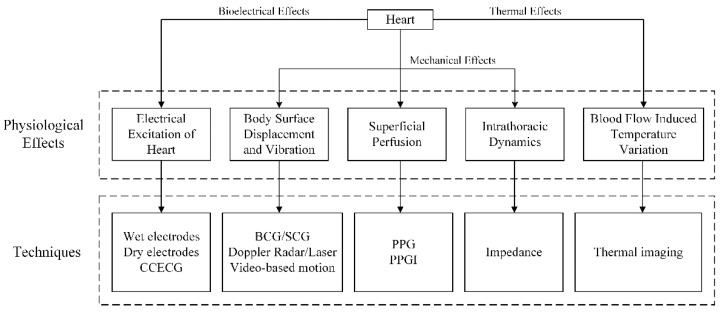
Overview of physiological effects and respective techniques for heart rate (HR) measurement.

**Table 1 sensors-20-02291-t001:** Summary of electrocardiography (ECG)-based techniques for HR monitoring.

Citation	Technique	Implementation	Movement	Power Consumption	Quantitative Result
Wu and Zhang [19]	CCECG	Integrated into a bedsheet	Sleep	NA	Root mean square error (RMSE): 0.66 ± 0.57 bpm
Gargiulo et al. [20]	Dry electrodes	Integrated into a chest strap	Exercise	33 mA (including transmission)	Correlation: larger than 0.96
Nemati et al. [10]	CCECG	Integrated into a stretchable cloth	Motionless	less than 25 mA	NA
Chen et al. [21]	Flexible dry electrodes	Integrated into a wrist band	NA	84.83 mW	NA
Rawstorn et al. [22]	CCECG	Integrated into a harness (chest strap)	Exercise	NA	Mean bias: −0.30 ± 4.53 bpm (sinus rhythm) 1.10 ± 9.75 (atrial fibrillation)
Dai et al. [23]	Flexible dry electrodes	Integrated into a garment	Sitting	29.74 mW	Accuracy: 98.55%
Dionisi et al. [24]	CCECG	Integrated into a T-shirt	Walking	17 mW (flexible solar panel)	Mean bias: 0.38 bpm
Zheng et al. [18]	CCECG	Integrated into chest strap	Exercise	2.1 mA	Mean bias: 0.60 ± 1.48 bpm
Li and Kim [25]	Dry electrodes	Integrated into a patch	Exercise	NA	Error rate: within 2% Correlation: 0.97

NA: not available. Movement: movement status of the subjects when HR measuring.

**Table 2 sensors-20-02291-t002:** Summary of finger-type photoplethysmography (PPG) sensors for HR monitoring.

Citation	Mode	Light Wavelength	Movement	Power Consumption	Quantitative Results
Rhee et al. [43]	Reflection	Red and Infrared	Shaking finger	Total current consumption: 0.491 mA; RF Transmitter: 0.098 mA; CPU-LED circuit: 0.393 mA	RMSE: 1.23 bpm
Maria Lopez-Silva et al. [44]	Transmission	Near infrared (850 nm)	Exercise	NA	μ: −0.7 bpm σ: 2.92 bpm LOA: [−6.41, 5.01] bpm
Park et al. [31]	Reflection	Red and Infrared	Motionless	Transmit mode: 31 mA; Receiving mode: 26 mA	NA
Yousefi et al. [39]	Transmission	Red (660 nm) and Infrared (895 nm)	Exercise	NA	μ: −0.57 bpm σ: 3.30 bpm LOA: [−7.0, 5.9] bpm

NA: not available. Movement: movement status of the subjects when HR measuring.

**Table 3 sensors-20-02291-t003:** Summary of ear-worn and patch PPG sensors for HR monitoring.

Citation	Sensor	Mode	Light Wavelength	Movement	Quantitative Results
Wang and Zheng [35]	Ear-worn	Reflection	Infrared	Motionless	RMSE: 1.3 bpm μ: −0.25 bpm σ: 0.5 bpm LOA: [−1.23, 0.73] bpm
Shin et al. [45]		Transmission	Infrared (940 nm)	Exercise	Error rates: 0.6% (rest); 1.7% (walk); 0.7% (jog); 5.7% (run)
Poh et al. [32]		Reflection	Infrared (940 nm)	Exercise	Stand: μ: 0.62%; σ: 4.51%; LOA: [−8.23, 9.46]% Walk: μ: −0.49%; σ: 8.65%; LOA: [−17.39, 16.42]% Run: μ: −0.32%; σ: 10.63%; LOA: [−21.15, 20.52]%
Poh et al. [46]		Reflection	Infrared	Exercise	Stand: μ: −0.07 bpm; σ: 2.56 bpm; Cycle: μ: −0.67 bpm; σ: 2.34 bpm; Walk: μ: 0.51 bpm; σ: 5.31 bpm;
Leboeuf et al. [47]		Reflection	Infrared	Exercise	μ: −0.2% σ: 4.4%
Alzahrani et al. [40]	Patch	Reflection	Green (525 nm) Red (650 nm) IR (870 nm)	Exercise	μ: 0.85 bpm σ: 9.20 bpm

NA: not available. Movement: movement status of the subjects when HR measuring.

**Table 4 sensors-20-02291-t004:** Summary of wrist-type PPG techniques for HR monitoring, part 1.

Method and Citation	Signal Processing Techniques	Error 1 (Mean ± SD) (bpm)	Error 2 (Mean ± SD) (%)	Bland-Altman Analysis (bpm)	Pearson Correlation Coefficient
SPECTRAP Sun and Zhang [54]	Spectrum subtraction based on asymmetric least squares	1.50 ± 1.95	1.12 ± 1.47	LOA: [−5.59, 6.01]	0.995
TROIKA Zhang et al. [52]	Sparse signal reconstruction: single measurement vector (SMV)	2.34 ± 0.82	1.80	μ: −1.24 σ: 3.07 LOA: [−7.26, 4.79]	0.992
JOSS Zhang [55]	Joint sparse spectrum reconstruction: multiple measurement vector (MMV)	1.28 ± 2.61	1.01 ± 2.29	LOA: [−5.94, 5.41]	0.993
MICROST Zhu et al. [56]	Wavelet and time-domain methods	2.58 ± 2.70	1.85	σ: 3.73 LOA: [−7.31, 7.31]	0.988
SpaMA Salehizadeh et al. [57]	Time-varying spectral filtering algorithm	0.89 ± 0.6	0.65 ± 0.4	NA	0.98
IMAT Mashhadi et al. [58]	Sparse reconstruction: iterative method with adaptive thresholding	1.25	NA	NA	NA

NA: not available.

**Table 5 sensors-20-02291-t005:** Summary of wrist-type PPG techniques for HR monitoring, part 2.

Method and Citation	Signal Processing Techniques	Error 1 (Mean ± SD) (bpm)	Error 2 (Mean ± SD) (%)	Bland-Altman Analysis (bpm)	Pearson Correlation Coefficient
FEEMD Zhang et al. [59]	Fast ensemble empirical mode decomposition (FEEMD) and spectrum subtraction	1.83 ± 1.21	1.4	σ: 3.62 LOA: [−7.56, 6.61]	0.989
MC-SMD Xiong et al. [60]	Multi-channel spectral matrix decomposition (MC-SMD) model	1.11	0.80	μ: 0.2248 σ: 1.9940 LOA: [−3.68, 4.13]	0.9968
EEMD Khan et al. [61]	Ensemble empirical mode decomposition (EEMD)	1.02 ± 1.79	0.79	LOA: [−4.10, 3.98]	0.996
Mix-SVM Xiong et al. [62]	Principle component analysis (PCA) and adaptive filter Sparse signal reconstruction Support vector machine (SVM) spectral analysis	1.01	0.72	LOA: [−3.46, 3.83]	0.9972
WFPV Temko [63]	Wiener filter and phase vocoder	1.02	0.81	NA	0.997
MURAD Chowdhury et al. [64]	Multiple reference RLS adaptive noise cancellation	0.9726 ± 1.831	0.76 ± 1.5	LOA: [−3.5665, 3.6112]	0.9972

NA: not available.

**Table 6 sensors-20-02291-t006:** Summary of photoplethysmographic imaging (PPGI) techniques for HR monitoring, part 1.

Citation	Sensor	Illumination	Distance (m)	Movement	Signal Processing Technique	RMSE (bpm)	Bland-Altman Analysis (bpm)	Pearson Correlation Coefficient
Poh et al. [72]	Webcam	Ambient light	0.5	Slight motion (sitting)	Independent component analysis (ICA)	Sitting still: 2.29 Slight motion: 4.63	Sitting sill: μ: −0.05; σ: 2.29 LOA: [−4.55, 4.44] Slight motion: μ: 0.64; σ: 4.59 LOA: [−8.35, 4.63]	Sitting still: 0.98 Slight motion: 0.95
Poh et al. [68]	Webcam	Ambient light	0.5	Motionless	ICA	1.24	NA	1
Sun et al. [73]	Monochrome CMOS camera	IR (870 nm)	0.4	Motionless	Planar motion compensation and blind source separation	NA	μ: 0.33 LOA: [−1.29, 1.96]	>0.9
de Haan and Jeanne [74]	CCD camera	Ambient light	NA	Cycling	Chrominance-based methods	0.4	NA	1
Holton et al. [75]	Webcam	Ambient light	0.6	Motionless	ICA	6.92	Standard error: 6.51 bpm	0.89

NA: not available. Movement: movement status of the subjects when HR measuring.

**Table 7 sensors-20-02291-t007:** Summary of PPGI techniques for HR monitoring, part 2.

Citation	Sensor	Illumination	Distance (m)	Movement	Signal Processing Technique	RMSE (bpm)	Bland-Altman Analysis (bpm)	Pearson Correlation Coefficient
Bousefsaf et al. [76]	Webcam	Ambient light	1	Head movements	Continuous wavelet filtering	2.33 ± 0.73	μ: 0.02 LOA: [−4.96, 4.99]	0.853 ± 0.056
Monkaresi et al. [77]	Webcam	Ambient light	NA	Cycling	Machine learning approach	4.33	μ: −0.28 σ: 4.33	0.97
Veeraraghavan et al. [78]	Camera	Ambient light	0.5	Facial movements	Combining skin-color change signals from different facial regions using a weighted average	NA	μ: 0.48 LOA: [−5.73, 6.70]	NA
Yu et al. [79]	Camera	Ambient light	0.6	Cycling	ICA	1.97	NA	0.99
Amelard et al. [80]	Monochrome camera	NIR	1.5	Supine position	Spectral-spatial fusion model	NA	µ: −1.0 σ: 0.70	0.9952

NA: not available. Movement: movement status of the subjects when HR measuring.

**Table 8 sensors-20-02291-t008:** Summary of PPGI techniques for HR monitoring, part 3.

Citation	Sensor	Illumination	Distance (m)	Movement	Signal Processing Technique	RMSE (bpm)	Bland-Altman Analysis (bpm)	Pearson Correlation Coefficient
Cheng et al. [81]	Webcam	Ambient light	0.5	Motionless	Joint blind source separation and ensemble empirical mode decomposition (JBSS–EEMD)	NA	μ: 1.15 σ: 8.46 LOA: [−15.43, 17.73]	0.53
Qi et al. [82]	Webcam	Ambient light	NA	Motionless	Joint blind source separation	5.0017	NA	0.7423
Bousefsaf et al. [83]	Webcam	Ambient light	1	Motionless	Segmentation based on lightness criteria	4.81	μ: 0.16 LOA: [−10.95, 11.26]	0.78
Tayibnapis et al. [84]	Webcam	Ambient light	0.3–1.1	Motionless	singular value Decomposition and Burg algorithm	3.34	μ: 2.15 σ: 2.58	0.73
Ling et al. [85]	Camera	Ambient light	0.6	Cycling	Canonical component analysis	Experiment 1: 3.70 Experiment 2: 2.33	NA	Experiment 1: 0.97 Experiment 2: 0.99

NA: not available. Movement: movement status of the subjects when HR measuring.

**Table 9 sensors-20-02291-t009:** Summary of Ballistocardiography (BCG) and Seismocardiography (SCG) measurements for HR monitoring.

Citation	Sensor	Movement	Quantitative Result
Wang et al. [94]	Pressure sensor	NA	Accuracy: 98.22%
Aubert and Brauers [103]	Electromechanical film sensors	Supine	Error: 1.25 bpm
Paalasmaa et al. [104]	Flexible piezoelectric film	Sleep	Mean absolute error: 0.78 bpm
Park et al. [105]	Piezoelectric film	Motionless	Standard deviation: 1.82 bpm
Bruser et al. [98]	Strain gauge	NA	Mean error: 0.39 bpm
Bruser et al. [97]	Strain gauge	Supine	Mean error: 0.46 bpm (10 s); 0.5 bpm (30 s)
Hernandez et al. [102]	Accelerometer, gyroscope, camera	Motionless	(gyroscope) Mean absolute error: 0.83 bpm
Tadi et al. [101]	Accelerometer	Supine	Average RMSE error: 0.33 bpm (supine); 0.62 bpm (right lateral); 0.45 bpm (left lateral)

NA: not available. Movement: movement status of the subjects when HR measuring.

**Table 10 sensors-20-02291-t010:** Summary of Doppler radar and laser measurements for HR monitoring.

Method	Device/Sensor	Distance (m)	Movement	Quantitative Result
Xiao et al. [108,109]	Ka-band Doppler radar	2	Motionless	Accuracy: 0.5 m, 100%; 1 m, 96%; 1.5 m, 89.3%; 2 m, 81.5%; 2.5 m, 64.6%
Xiao et al. [110]		2.8	Motionless	Accuracy: 0.5, 1, 1.5, 2, 2.8 m: 98.82%, 91.71%, 92.40%, 85.78%, 81.35%
Li et al. [111]		0.5–2.5	Motionless	Accuracy: 0.5 m, 1 m, 1.5 m, 2 m, 2.5 m: 99.1%, 89.8%, 98.9%, 85.2%, 83.3% (front); 96.3%, 89.8%, 89%, 80.5%, 85.7% (left); 100%, 93.2%, 93.8%, 97.4%, 85.1% (right); 97.6%, 100%, 94.3%, 93.6%, 85.5% (back)
Tavakolian et al. [112]	Doppler radar	0.1	Motionless	Accuracy: 92.9%
Obeid et al. [113]		NA	Motionless	Relative error: 0.5–1.5%
Morbiducci et al. [114]	Laser Doppler vibrometer	1.5	Motionless	Bias: 0.006 bpm (male);0.015 bpm (female)
Scalise and Morbiducci [107]		1.5	Motionless	Mean bias: 0.026 bpm

NA: not available. Movement: movement status of the subjects when HR measuring.

**Table 11 sensors-20-02291-t011:** Comparison of different HR monitoring techniques.

Technique	Measuring Sensor	Distance	Movement	Cost
PPG	Phototransistor	mm	Exercise	low
PPGI	Camera/webcam	m	Motionless	low
Thermal imaging	Thermal imaging camera	m	Motionless	highest
BCG/SCG	Pressure sensor, strain gauge, optical sensor, etc.	mm	Motionless	low
Video-based motion	Camera/webcam	m	Motionless	low
Radar	Microwave sensor	m	Motionless	medium
Laser	Laser	m	Motionless	high
Wet ECG	Wet electrodes	0	Subtle Motion	medium
Dry ECG	Dry electrodes	0	Exercise	medium
CCECG	Capacitively coupled electrodes	mm	Exercise	medium
Impedance	Coils/electrodes	cm	Motionless	medium

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
