# Peer review of "Is Continuous Heart Rate Monitoring of Livestock a Dream or Is It Realistic? A Review"

_sensors, 2020, doi:10.3390/s20082291_

Round 1

Reviewer 1 Report

Summary:

This paper starts by presenting state-of-the-art techniques for heart rate (HR) monitoring (mainly in humans) and lists their advantages and drawbacks. The challenges to transfer these technologies to livestock are then described. The authors chose the PPG technique as the most promising technique and list the main challenges related to the use of PPG in livestock. The paper is well written and has appropriated references to the previous work.

Broad comments:

The paper gives the impression that the authors choose the PPG technique before writing the article. The abstract is also written in the sense that PPG is your best candidate. I wonder if you shouldn’t change the title of the paper to underline that you evaluate the use of PPG for HR monitoring of livestock and compare it to other state-of-the-art techniques.

The word “continuous” in the abstract (line 19) is misleading. It let the reader think that the HR is continuously measured, which is not the case for all the listed technologies. Especially for the PPG (which is the central technology of the article) where the system can be few minutes in “sleep” mode between two measurements (to save energy). Please be careful with the use of this word in the article (also in the title).

HR monitoring with the PPG technique during intensive physical exercise is a major challenge (as reported in the submitted paper). However, there is a major difference between rhythmic (e.g., walking, running) and non-rhythmic motion artifacts (as it is the case for livestock). In my opinion, adding a paragraph in the paper to underline this important aspect of PPG signal processing is necessary (or reworking the “Motion artifacts removal” paragraph of section 5).

Bioimpedance can also be used to measure HR via superficial/internal blood perfusion (mechanical effect). To be more exhaustive, it could be mentioned in the paper and explain why this “candidate” is not appropriate for livestock application.

Specific comments:

Line 17: “recently” is maybe true for PPG HR monitors, however ECG based system exists for many years (ECG holters, polar belt, etc.)

Line 33: FAO abbreviation is defined only in reference [1], please add the definition in the text and/or in the abbreviations list.

Line 89: groups (1) and (2) were subcategories of mechanical effect. Remove the (3) for the thermal effect to be compliant with figure 1.

Line 92, Figure 1: Add bioimpedance technique to be more exhaustive (c.f., abovementioned broad comment).

Line 148: the word “eliminates” imply that the sensor placement is not a problem anymore, please reformulate the sentence. Maybe “reduces” would be more appropriate.

Line 159: replace “strong” by a more specific adjective. Maybe “high amplitude” would be more appropriate.

Line 165: “excellent” is a personal point of view. Even if this is an excellent review, be careful to remain objective.

Line 240: Does the blood flow via the abdominal aorta to go from the heart to the head? Maybe the wrong word was used.

Line 276: a mechanical contact is still required, isn’t it?

Table 11: How do you justify that PPG sensor is cheaper than ECG systems?

Line 362: “MAs” instead of “Mas”.

Line 540: please use the correct unit: 80 mAh. Moreover, 80 mAh divided by 26 mA results in an autonomy of about 3 hours. This mean that the referred system does not work in continuous time. Please make it clear in the text.

Line 595: Abbreviations should be sort in alphabetic order. Some abbreviations are also missing.

Reviewer 2 Report

The authors proposed an interesting and exhaustive review of different techniques to measure HR on living organisms. In the last years, protection and animal welfare covered an important role. There are a lot of investments in this field and it appears that this is important sustainability and welfare topic and the comparative techniques used to measure the HR reported in this paper could have a significant and positive impact on the farming and biomedical community. Overall, the introduction, the techniques for monitoring the HR are quite well done. Moreover, the part dedicated to these techniques applied to the livestock it was rich and comprehensive.
In general, the manuscript is reasonably well done and it ready for future publication.

The authors, in the future, should also consider adding some reports or papers about some international policies adopted in this field in the last year from the big countries(An example, the European Commission has been promoting animal welfare for over 40 years gradually improving the lives of farm animals).

Reviewer 3 Report

In my opinion, the paper is very good and can be accepted with only small changes in editing. Some small things such as some abbreviations need to define before using them such as PPG, FAO. Authors should clearly explain their solutions to deal with the continuous HR sensor for livestock.

Round 2

Reviewer 1 Report

The authors have adequately answered to the comments and clarified the key points in the manuscript.